# Electron scale coherent structure as micro accelerator in the Earth's magnetosheath

Zi-Kang Xie [1], Qiu-Gang Zong [1,2] ✉, Chao Yue [1], Xu-Zhi Zhou [1], Zhi-Yang Liu [1], Jian-Sen He [1], Yi-Xin Hao[3], Chung-Sang Ng [4], Hui Zhang[5], Shu-Tao Yao[5], Craig Pollock[6], Guan Le[7], Robert Ergun[8] & Per-Arne Lindqvist [9]

Turbulent energy dissipation is a fundamental process in plasma physics that has not been settled. It is generally believed that the turbulent energy is dissipated at electron scales leading to electron energization in magnetized plasmas. Here, we propose a micro accelerator which could transform electrons from isotropic distribution to trapped, and then to stream (Strahl) distribution. From the MMS observations of an electron-scale coherent structure in the dayside magnetosheath, we identify an electron flux enhancement region in this structure collocated with an increase of magnetic field strength, which is also closely associated with a non-zero parallel electric field. We propose a trapping model considering a field-aligned electric potential together with the mirror force. The results are consistent with the observed electron fluxes from ~50 eV to ~200 eV. It further demonstrates that bidirectional electron jets can be formed by the hourglass-like magnetic configuration of the structure.

Plasma turbulence is one of the fundamental physical phenomena that has not been fully understood, as it is complex in energy transfer from large to small scales and in energy conversion between fields and particles[1,2]. A series of evidence shows that it may exist throughout the universe, such as early universe[3], crab pulsar[4], interstellar medium[5], planetary magnetosphere[6], etc. It is thought that plasma turbulence may play a key role in particle energization, such as solar corona heating[7] and cosmic ray acceleration[8]. Links to other basic plasma physical processes are widely investigated, such as magnetic reconnection[9], ring current[10], etc. A main challenge in turbulence research is the multi-scale coupling. However, large-scale physical processes can be well described by magnetohydrodynamic (MHD) theory[11–13]; the coupling between plasma kinetics and the ultimate scale of dissipation is still not clear[14–16].

Coherent structures can be formed self-consistently in turbulence with an inhomogeneous distribution of energy transfer. There are various types of coherent structures in plasma turbulence, e.g., vortices and current sheets. Tremendous efforts have been made to search for coherent structures in space plasmas[17], laboratory plasmas[18,19], and numerical plasmas[20]. Various conditional sampling methods[21] have been also developed to identify coherent structures, in which the partial variance of increments (PVI) method was recently developed and widely used[22,23].

It is thought that these structures have a direct connection to turbulent energy cascading and dissipating mechanisms such as the anomalous transport[24], and the scale length may vary from large scale[25] to kinetic size[26]. Intermittent coherent structures with stronger current density, especially for the first class, are usually associated with enhancements in temperature, indicating plasma heating due to dissipation of coherent structures[27,28]. Dissipation of both coherent currents and coherent vortices can be responsible for plasma energization. It is found from numerical simulations that local energy

[1]Institute of Space Physics and Applied Technology, Peking University, Beijing 100871, China. [2]State Key Laboratory of Lunar and Planetary Sciences, Macau University of Science and Technology, Taipa, Macau, China. [3]Max Planck Institute for Solar System Research, Göttingen, Germany. [4]Geophysical Institute, University of Alaska Fairbanks, Fairbanks, AK, USA. [5]Shandong Provincial Key Laboratory of Optical Astronomy and Solar-Terrestrial Environment, Institute of Space Sciences, Shandong University, Weihai 264209, China. [6]Denali Scientific, 3771 Mariposa Lane, Fairbanks, AK 99709, USA. [7]Heliophysics Science Division, NASA, Goddard Space Flight Center, Greenbelt, MD 20771, USA. [8]Department of Astrophysical and Planetary Sciences, University of Colorado LASP, Boulder, CO, USA. [9]Department of Space and Plasma Physics, KTH Royal Institute of Technology, Stockholm, Sweden. ✉e-mail: qgzong@pku.edu.cn

transfer rate of turbulence, electromagnetic field work ($\mathbf{J} \cdot \mathbf{E}$) related with current dissipation, pressure-strain interaction ($-(\mathbf{P} \cdot \nabla) \cdot \mathbf{u}$) related to vortex dissipation can be systematically converted into thermal energy in space where plasma temperature is locally enhanced[29–31].

The recent launch of the magnetospheric multiscale (MMS) mission[32] has brought the space exploration into electron kinetic scales. The turbulence in magnetosheath measured by MMS can be decomposed quantitatively into various wave modes from ion scales down to sub-electron scales: kinetic Alfvén waves, whistler waves, and ion acoustic waves[33]. The intermittency of electric field down to electron scales is explored for the first time, and is found to behave as strong multi-fractal, evidently different from the mono-fractal of magnetic field[34]. A series of electron-scale coherent structures have been identified and reported in space plasma environment such as electron-scale magnetic cavity[35–38], electron-scale current sheet[39], etc. Study on these structures may have special significance, as it is currently believed that turbulent energy may be finally dissipated at electron scales. Recent advances implied that the dissipation might occur through wave-particle interactions[40]. Evidences of Landau damping of kinetic Alfvén waves and cyclotron damping of ion cyclotron waves have been discovered together with the signatures of field-particle correlation as well as spectra of dissipation rate[41,42]. However, it still remains as an essential question in magnetized plasma on what the ultimate scale that the turbulent energy can cascade down to is.

Now, with NASA's MMS mission[32] of four identical satellites launched in March 2015, we are able to obtain multi-spacecraft observations of ion or even electron-scale electromagnetic structures due to its small separation (several to tens of kilometers) and high time resolution (millisecond). The mission provides high-time resolution magnetic field (FGM[43], 7.8 ms on burst mode), electric field (EDP[44,45], 30 ms on fast mode and 0.12 ms on burst mode), and plasma (FPI[46], 30 ms for electrons and 150 ms for protons on burst mode) measurements, creating unprecedented opportunities for the study of electron-scale coherent structures.

In this paper, we present MMS multi-point observations of an electron-scale coherent structure in turbulent terrestrial magnetosheath. The observation shows twisted magnetic field lines and trapped electrons in the peak region of magnetic field strength ($B_t$), accompanied by a non-zero parallel component of the electric field. We then develop a trapping model of electrons considering a field-aligned electric potential drop together with the mirror force and find that the electron trapping and acceleration could be well ascribed to the variation of electric potential along the field-aligned direction. Further analysis shows that a bidirectional electron jet is formed at the end of the structure by magnetic mirror force.

## Results

### Observations

The MMS four satellites were located in the Earth's magnetosheath but not far away from the magnetopause between 10:26:25 and 10:26:29 UT (universal time) on 21 Sep 2015.

Figure 1 shows MMS1 observations of the electromagnetic fields, plasma environment, and electron pitch angle distributions (PADs) across multiple energy channels as an overview of the coherent structure. The spatial separation between different MMS spacecraft is around 40 km, and the MMS location in geocentric solar ecliptic (GSE) coordinates is given at the bottom of this figure, i.e., (6.5, 8.8, −0.1) $R_E$ (Earth's radius, 6371 km). Figure 1a shows the plasma environment in the magnetosheath in a time scale of 3 min. As shown, the magnetic field in the magnetosheath is very turbulent. The coherent structure we are interested in is shown in Fig. 1b–l with a time scale of 1 s (line-1 to line-4).

The magnetic field (Fig. 1b, c) and the electron pressure tensor (Fig. 1e) are projected to a local field-aligned coordinate system.

The +z-axis is defined as a parallel direction based on the mean magnetic field over the four MMS spacecraft between 10:26:26 and 10:26:28 UT, and the +y-axis is defined by a mean perpendicular proton bulk velocity between 10:26:26 and 10:26:28 UT. Thus, the spacecraft motion in the perpendicular plane could be considered as in the −y direction. The x-axis completes the orthogonal set. The three newly defined axes of the local field-aligned coordinates are $\mathbf{e_x} = [0.84, 0.51, -0.16]$, $\mathbf{e_y} = [-0.54, 0.78, -0.33]$, $\mathbf{e_z} = [-0.04, 0.36, 0.93]$ in the GSE coordinate system.

Figure 1b, c shows a significant enhancement of the field strength $B_t$ between line-1 and line-4 and a clear bipolar variation of the perpendicular magnetic field (Fig. 1c), resembling a twisted magnetic field configuration at a very small-time scale (1 s). Figure 1d shows a slight depletion (-12%) of the electron number density simultaneously with the enhancement of $B_t$. Figure 1e shows the diagonal terms of the electron pressure tensor, which indicate a large depletion in the parallel direction (-40%) and a less prominent (-15%) change of the perpendicular components.

Figure 1f presents the parallel component of the electric field (only MMS1 observations are used), where line-1 represents the beginning of non-zero $E_\parallel$; line-2 and line-3 mark the time when the sign of $E_\parallel$ is reversed; line-4 is the end of non-zero $E_\parallel$. The $E_\parallel$ changes its sign at the time marked by line-2, implying that the electric potential reaches its maximum. Therefore, a single electron would undergo an acceleration process when traveling from line-1 (or line-3) to line-2, during which the electric potential energy is converted to kinetic energy. Figure 1h shows the $\mathbf{J} \cdot \mathbf{E}'$ inside the structure (only MMS1 observations are used, all interpolated to electron sampling time of FPI-DES, and it is assumed that the electron and ion number densities are equal). In line-2 and line-3, the energy transfer between fields and particles ($\mathbf{J} \cdot \mathbf{E}'$) implies that the field received energy at the center of the structure and released the energy at the end, ejecting electrons to the ambient plasma along the field line.

Figure 1i–l shows the electron pitch angle distributions in the energy channels from ~90 eV to ~200 eV. The non-zero parallel component of the electric field (~0.6 mV/m, panel (f)) provides a possibility that these electrons are trapped by a combination of parallel electric field and mirror force.

The overplotted magenta lines in Fig. 1i–l are the critical trapping angle $\alpha_t$, which is derived from Eq. (7) in the method section considering the existence of a parallel electric field. This equation illustrates that, unlike pure mirror trapping, $\alpha_t$ is energy-dependent due to the presence of the electric potential $\Phi$, which is key to the value of $\alpha_t$. Here, we give an estimation of $\Phi$ by comparing the actual observed pitch angle $\alpha$ in the electron fluxes enhancement region with the $\alpha_t$ derived from Eq. (7). Here, the background magnetic field strength is set to be 27.3 nT, and there is only mirror force to trap electrons before line-1. The maximum of $\Phi$ is estimated to be 50 V, as shown in Fig. 1g. Based on the observation shown in Fig. 1f that the $E_\parallel$ reversed its direction at the time marked by line-2 and line-3, we consider a linear increase from 0 to the maximum, then a linear decreasing from its maximum to 0, again a linear increasing to its previous maximum between line-1 and line-2, line-2 and line-3, line-3 and line-4, respectively.

### Electron jet at the end of the structure

It is worth mentioning that there is a bidirectional electron jet at the end of the structure, as is shown in Fig. 1i–l. The jet starts to form at line-3 where the electrons begin changing its dominant pitch angle from 90° to 45° or 135° (after line-3) at energies between ~50 eV and ~200 eV. Note that line-3 is the end of the electron trapping region where the inward electric force by $E_\parallel$ vanishes. If we regard line-3 as a source for leaked electrons (60°–120° pitch angle), we would be able to deduce the distribution after line-3 given a certain electromagnetic field configuration. The blue curves in Fig. 1i–l represent the deduced

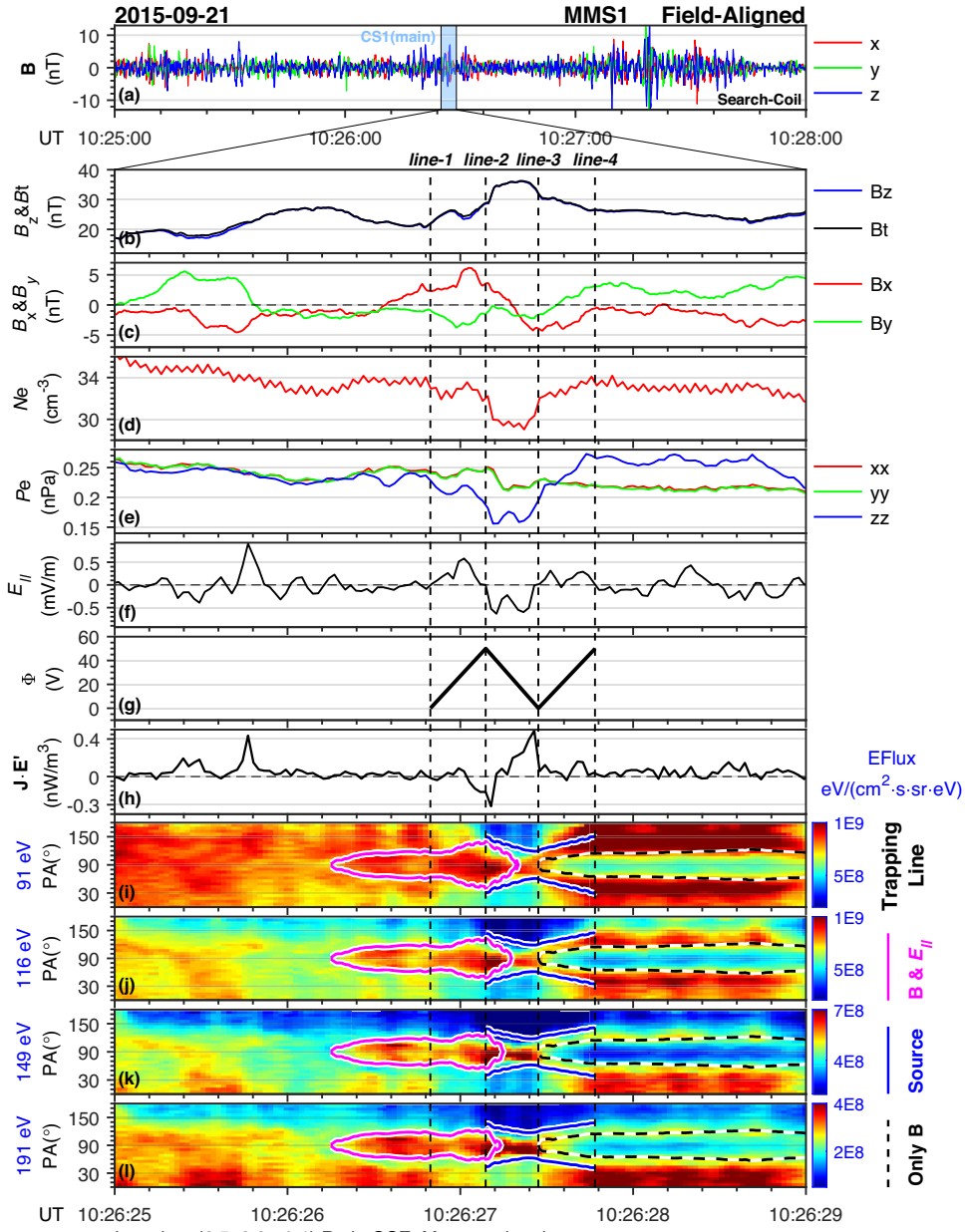

**Fig. 1 | MMS observations of an electron-scale coherent structure (CS) in turbulent magnetosheath.** Panel **a** shows the turbulent magnetic fields at a 3-min time scale, while the rest of panels b–l are for only 4 s (CS1). Panels b and c show magnetic field components ($B_x$ in red, $B_y$ in green, $B_z$ in blue, and the field strength $B_t$ in black) in the newly defined local field-aligned coordinates (details in the main text). Panels d–f denote electron number density, diagonal terms of electron pressure tensor, and parallel electric field ($E_\parallel$), respectively. Panel **g** shows the assumed electric potential detailed in the main text. Panel h shows the $\mathbf{J} \cdot \mathbf{E}'$. Panels i–l show pitch angle distributions of electron energy flux of energies from ~90 eV to ~200 eV. There are four vertical dashed lines across all panels, where line-1 represents the beginning of non-zero $E_\parallel$; line-2 and line-3 mark out the time when the sign of $E_\parallel$ is reversed; line-4 is the end of non-zero $E_\parallel$. In panels **i–l**, the magenta lines represent the critical trapping angle $\alpha_l$ defined by Eq. (6); the blue lines show the expected streaming region given an electron source at line-3 (60° to 120° pitch angle); the dashed black (white) lines reproduce single particle motion for an electron starting at line-3 (90° pitch angle) without the impact of $E_\parallel$. All the lines (magenta, blue, and dashed) are directly deduced by electromagnetic field observations (panels (b), (c), (f), and (g)). Spacecraft position is labeled at the bottom of the figure.

electron distributions taking the variation of $\Phi$ (Fig. 1g) into account together with $B_t$. It is illustrated that the results (blue curves) can have a good agreement with the outer edge of the electron jet, supporting the scenario that electrons leaked at line-3 are further pumped out by an outward electric force between line-3 and line-4, forming an accelerated electron jet at the end of the structure.

In Fig. 1i–l, the dashed black lines after line-3 are pitch angle variations of the 90° electrons at line-3 due to the decrease of $B_t$ after line-3 without considering an $E_\parallel$, i.e., the pure mirror effect as

demonstrated in Eq. (1). The dashed black lines can match the inner edge of the electron jet, which means that there are few locally trapped electrons. In other words, the main population is from the source at line-3 and undergoes the electric field acceleration.

**Trapped electrons within a strong magnetic field region**
Figure 2 shows the electron energy fluxes in velocity space at different times, where magenta lines represent the shape of $\alpha_l$ in $v_\parallel$ - $v_\perp$ coordinates. Two left panels (Fig. 2a, b) are observations at line-1 and line-2,

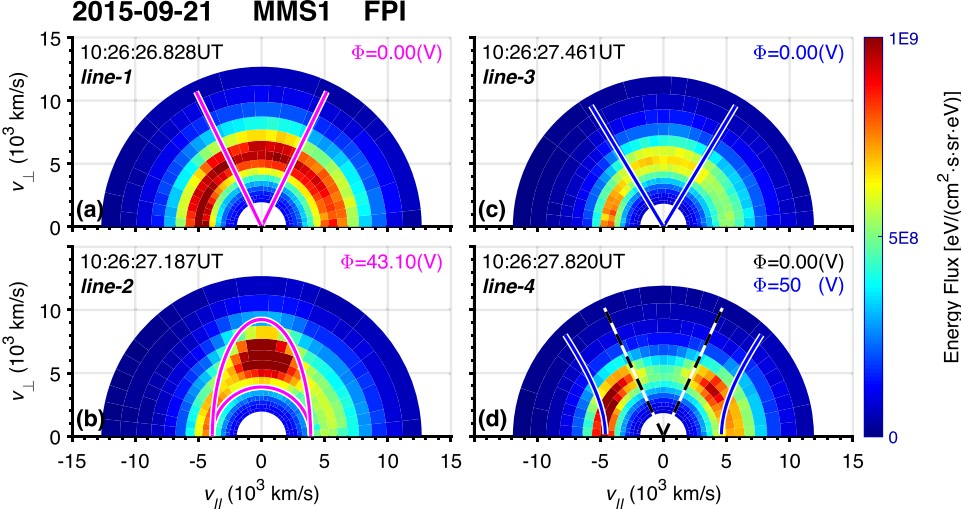

**Fig. 2 | Electron energy fluxes in $v_{\parallel}$–$v_{\perp}$ coordinates.** For all panels, the x- and y-axis are for $v_{\parallel}$ and $v_{\perp}$, respectively. In each panel, the universal time is labeled in the top left corner corresponding to one of the four vertical dashed lines in Fig. 2, and the assumed electric potential is labeled in the top right corner. The magenta line in panel (a) is the critical trapping angle $\alpha_l$ given by Eq. (6). In panel b; there is an additional magenta line (the semicircle) representing the cut-off kinetic energy limit $E_k = e\Phi$. The blue line in panel (c) is simply 60° and 120° pitch angle, the region between which is regarded as the source of the electron jet after line-3. The dashed black line in panel d illustrates the pitch angle limit due to purely mirror force (i.e., Eq. (4)). The blue line in panel d illustrates the critical trapping angle $\alpha_l$ given by Eq. (6).

respectively, in which the time point is labeled at the top left corner. Note that if there is only mirror force, the critical trapping angle $\alpha_l$ will be energy-independent, i.e., a straight line stretching out from (0,0) in $v_{\parallel}$ - $v_{\perp}$ coordinates as shown in Fig. 2a, demonstrating that mirror force makes the full contribution to $\alpha_l$ (since $\Phi$ is zero). The enhancement of electron energy fluxes disappears at higher energies simply due to much lower density. On the contrary, in Fig. 2b, the $\alpha_l$ are entirely different from straight lines due to a relatively high $\Phi$ value (43.10 V). The half-circle shape of the trapping limit represents a cut-off kinetic energy $E_k = e\Phi$, and the other limit stretching out to higher energies represents those $\alpha_l$ values derived from Eq. (7). In addition, the shape of trapping angle $\alpha_l$ coincides well with the enhancement of electron fluxes, including the upper boundary representing $\alpha_l$ of Eq. (7) and the lower boundary for $E_k = e\Phi$, suggesting a significant electric field impact on the structure. Actually, the closed region surrounded by the magenta curve represents the particles trapped by the magnetic and parallel electric fields. In contrast, outside the closed region, it means the composition of the particles passing through freely. Since $B_t$ is strong at line-2, the mirror force would push electrons outward away from that region, and the electrons could only be trapped due to the presence of the electric force. The highest energy that electrons are trapped is about 200 eV in Fig. 2b, indicating that the mirror force would dominate the motion of electrons with higher energies (the mirror force $\mathbf{F} = -E_k \cdot \nabla_{\parallel}B/B$ is proportional to kinetic energy $E_k$).

The source distribution is shown in Fig. 2c, in which the blue lines are simply 60° and 120° pitch angles, and the region between these two blue lines is regarded as the source of the electron jet after line-3. The jet distribution is shown in Fig. 2d, in which the dashed black line illustrates the pitch angle limit due to purely mirror force described by Eq. (1). As shown, the observed electron jet distribution is consistent with the theory prediction due to the magnetic mirror force.

**Scale size evaluated by an energetic particle sounding technique**
It has been illustrated that the energetic particle sounding technique is applicable to burst mode electron phase space density (PSD) data from MMS FPI instruments[37]. For this event, we also find that there are clear electron non-gyrotropic distributions inside the structure, as shown in Fig. S1 in the Supplementary material. Note that the electron fluxes of 90° pitch angle significantly decrease between line-3 and line-4, which

causes a relatively large uncertainty for sounding due to insufficient counts. Thus, the electron distributions of 60° pitch angle are used for sounding.

Figure 3c−e shows the results of the sounding technique, and Fig. 3a, b are $E_{\parallel}$ and electron PAD of 116 keV for reference. In panels (d) and (e), the orientations are represented by the combinations of azimuthal angles and polar angles in the newly defined local field-aligned coordinates, which shows consistency across multi-energy channels. The polar angle remains close to 120°, while the azimuthal angle is around zero when the electron pitch angle is 60°.

The distances away from the trapping boundary shown in Fig. 3c demonstrates good agreements across multi-energy channels between line-1 and line-4, suggesting that the trapping boundary for the four energy channels are the same and the sounding technique is applicable for this structure. The average distance of the boundary is around 2 km, which is equal to 2.2$\rho_e$ (The local electron cyclotron radius $\rho_e$ is ~0.9 km), indicating that the structure is at the electron scale. Note that the 90 and 116 eV channels are saturated between line-3 and line-4 since the distance is larger than twice of the gyro-radii. The distance (~1 km) between line-2 and line-3 is significantly smaller than that between line-1 and line-2 or line-3 and line-4, implying that in the $B_t$ maximum region, the diameter of the cross-section is reduced.

## Discussion
In this study, we have investigated a coherent structure in the terrestrial magnetosheath and found clear flux enhancement of electron PADs (between lines 1−3 of Fig. 1) when $B_t$ reaches its maximum, which cannot be explained by mirror trapping. We then suggest a trapping model considering an electric potential $\Phi$ along the field-aligned direction together with the mirror force, demonstrated by Eqs. (1)–(7) in the method section, based on observations of non-zero $E_{\parallel}$ in Fig. 1f. By assuming a linear increasing $\Phi$ from 0 V to 50 V in positive $E_{\parallel}$ region between (line-1, line-2) and (line-3, line-4), and a linear decreasing $\Phi$ from 50 V to 0 V in negative $E_{\parallel}$ region between (line-2, line-3) as shown in Fig. 1g, this model is able to give a critical trapping pitch angle $\alpha_l$ derived from Eq. (7), which is in good agreement with the electron flux enhancement shown in panels Fig. 1i–k. In Fig. 2, we further demonstrate the consistency between $\alpha_l$ and the electron flux observations in velocity space. The electrons might be accelerated in the middle of the

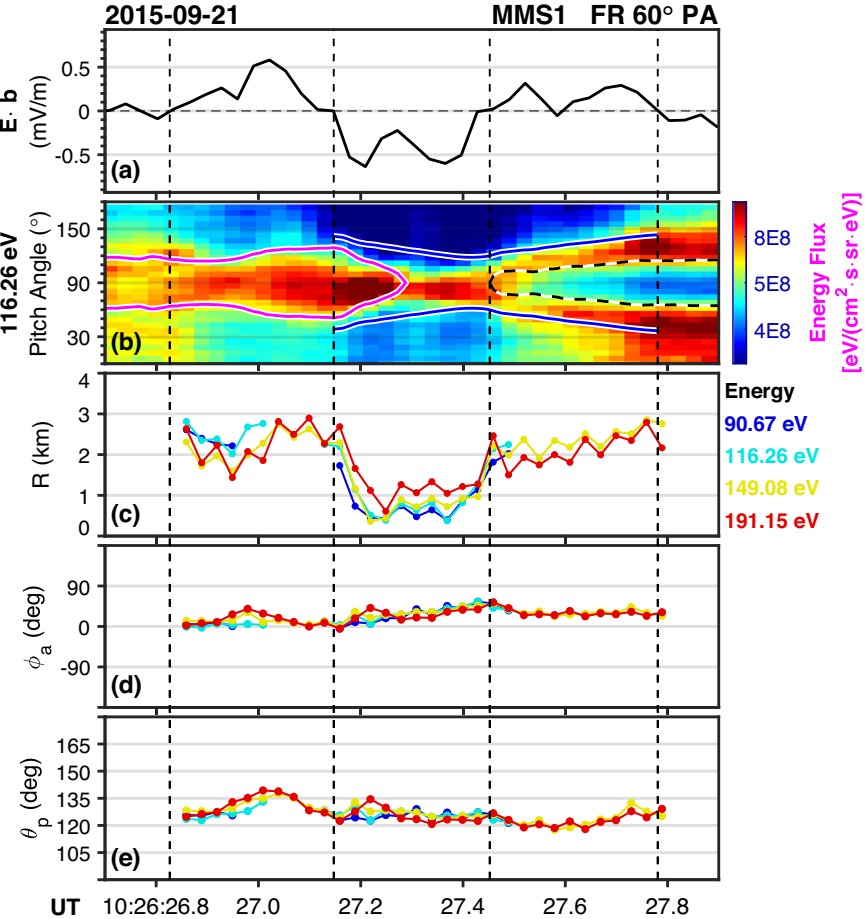

**Fig. 3 | Boundary orientations and distances derived from the energetic particle sounding technique.** Panels (a) and (b) are $E_\parallel$ and electron PAD for a specific energy channel as an example (previously shown in Fig. 2), respectively. Panel (c) shows the variation of the distances ($R$) at four energy channels. Panels (d) and (e) are orientations ($\phi_a$ for azimuthal angle and $\theta_p$ for polar angle) of the boundary in the newly defined local field-aligned coordinates. Four vertical dashed lines are the same as in Fig. 2.

structure by the $E_\parallel$. Along with the parallel electric field in the structure, there is energy transfer between fields and particles.

The scale size is also estimated by using a sounding technique. It is shown that the average distance of the trapping boundary to the spacecraft may be approximately 2 km, i.e., $2.2\rho_e$. However, the variation of the distance shown in Fig. 3c implies that the boundary may be significantly curved. As shown in Fig. 3c, the structure scale is large at both ends and small in the middle, which may mean the structure is most likely to resemble an hourglass shape. This shape resembles an hourglass-like picture, as shown in Fig. 4a, where the diameter of the cross-section at the center is significantly smaller than the outer part. The continuous decrease of $B_t$ as well as the observed electron jet after line-3 (Fig. 1) further supports this scenario.

As shown in the schematic diagram of Fig. 4a, the hourglass-like structure can trap electrons between ~50 eV to ~200 eV in the center of the structure by the combined effect of a non-zero parallel electric field and the magnetic mirror force $\mathbf{F} = -E_k \cdot \nabla_\parallel B/B$. The shape of twisted magnetic field lines in Fig. 4a is inferred from MMS observations shown in Fig. 1b, c. The streaming electrons (Fig. 1i–l after line-3) are labeled at the end of the structure in Fig. 4a. The total potential $\Phi_{\text{total}}$ (red line), which is the sum of non-zero parallel electric potential $-e\Phi_\parallel$ (green line) and the potential $\Phi_{-\mu\nabla_\parallel B}$ (blue line) generated by the gradient force of the magnetic field in the parallel direction shown in Fig. 4b qualitatively, contribute to the electron trapping (peak island region) at the center of the structure.

Boldyrev et al. discussed electron trapping and bidirectional electron jets along a magnetic field if the field strength is continuously decreasing and an electric potential variation is presented[47]. It is assumed that there is an isotropic electron source at $r_0$ (where the magnetic field strength is $B_0$ and the electric potential is zero) so that at a new position $r$ (where the magnetic field strength is $B(r)$ and the electric potential energy is $e\Phi(r)$), there are streaming electrons from the source[47]. The population at $r$ is then divided into three parts[47]: (1) streaming electrons, from the source but will bounce back at some point; (2) runaway electrons from the source but are able to reach infinity where $B(r) = 0$ and $e\Phi(r) = e\Phi_\infty$; (3) locally trapped electrons, that do not come from the source. The Critical trapped line turns out to be a similar elliptic curve, as shown in Fig. 2b.

Although the discussion based on Boldyrev et al. is not directly applied to the electron trapping in strong $B_t$ region, it can explain the bidirectional electron jet after line-3 in Fig. 1i–l. Given that electrons can bounce back during the motion along the field line, the bidirectional electron jet is formed due to the source at line-3 (leaked electrons). It is further illustrated in Supplementary Fig. 2 that the intensity of the inward jet is close to the outward jet, suggesting that runaway electrons are not prominent in this structure. The result shows that the electron-scale coherent structure discussed here can be an electron accelerator and an electron jet driver, which may have a link to the formation of strahl electrons in the solar wind. The process from isotropic electrons to bidirectional jets is also sketched in Fig. 4a.

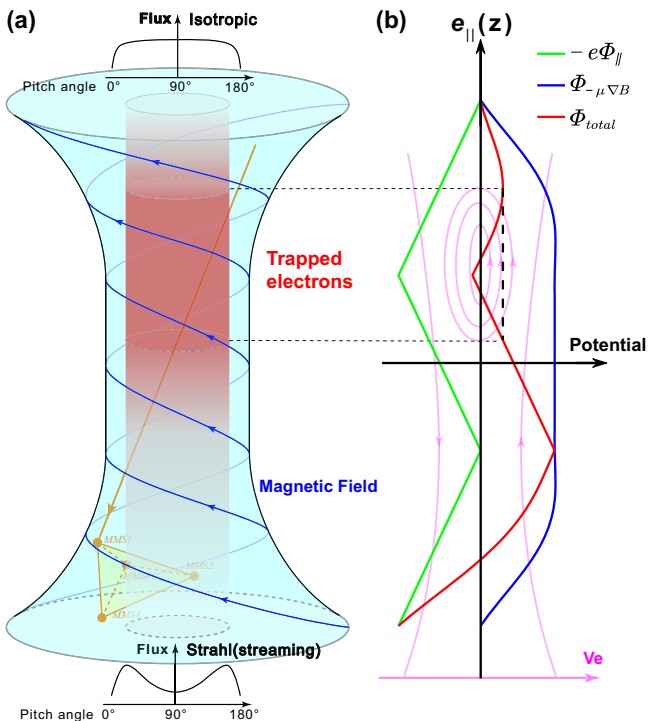

**Fig. 4 | Schematic diagram of the hourglass electron-scale coherent structure with trapped electrons and non-zero parallel electric fields. a** Sketch of the hourglass structure. The blue curve represents twisted magnetic fields with a cylindrical symmetric schema. Spacecraft trajectory (MMS1) is labeled as an inclined upward dark orange line. **b** The non-zero parallel electric potential $-e\Phi_{\parallel}$, the potential $\Phi_{-\mu\nabla_{\parallel}B}$ generated by the gradient force of the magnetic field in the parallel direction, and the total potential $\Phi_{total}$ are labeled with green, blue, and red lines, respectively. The total potential contributes to the electron trapping (red region) at the center of the structure. The scale size of the structure is estimated to be at the electron scale. The pink island represents the possible orbits of trapped and runaway particles in phase space.

There are various types of coherent structures in turbulent plasmas in microscale. Supplementary Figs. 3 and 4 show two extra structures in the magnetosheath, which show similar features as the case demonstrated in Fig. 1. More similar coherent structures can be found in Supplementary Table 1. In summary, we propose an electron-scale coherent structure that has been found in a turbulent environment, as schematically illustrated in Fig. 4. In the model, electrons are trapped and accelerated at the center of the structure by a bipolar parallel electric field in a $B_t$ maximum region. At the end of the structure, a bidirectional electron jet is formed due to an outward parallel electric force together with the outward mirror force, which further accelerates electrons and impacts on electron dynamics in the ambient plasma.

## Methods
### Trapping model
It is natural that particles can be trapped in a local magnetic bottle by mirror force, where the enhancement region of PADs can have a good agreement with the variation of critical trapping angle[48]:

$$\alpha_l^* = \arcsin\left(\sqrt{\frac{B}{B_t}}\right) \tag{1}$$

where $B_t$ is background magnetic field strength. Similarly, bidirectional electric force along a field-aligned direction can also contribute to particle trapping, thus, Eq. (1) can be modified by introducing an electric potential. In the potential field induced by the parallel electric

force, the particle's total energy $W$ remains constant. Recalling $\Phi(s)$ denoting the potential, we have:

$$W = E_k(s) + q\Phi(s) = E_{k0} + q\Phi_0 = \text{const} \tag{2}$$

Assume that the initial potential is 0 (The 0 subscript represents the area with no electric field), $\Phi_0 = 0$, we can get:

$$E_{k0} = E_k(s) + q\Phi(s) \tag{3}$$

We next assume that the first adiabatic invariant $M$ is conserved:

$$M = \frac{E_{k\perp}}{B} = \frac{E_k \sin^2(\alpha)}{B} = \text{const} \tag{4}$$

Assuming that the background magnetic field strength outside the structure is $B_t$ and substituting Eq. (3) into (4), we have,

$$M = \frac{E_k \sin^2(\alpha)}{B} = \frac{E_{k0} \sin^2(\alpha_0)}{B_t} = \frac{(E_k + q\Phi)\sin^2(\alpha_0)}{B_t} \tag{5}$$

Accordingly, the pitch angle $\alpha$ of a particle entering into the structure would be:

$$\sin(\alpha) = \sqrt{\frac{B}{B_t} \cdot \frac{E_k + q\Phi}{E_k}} \sin(\alpha_0) \tag{6}$$

where $B$ and $\Phi$ is the magnetic field strength and the electric potential at the particle position, respectively; $E_k$ is the kinetic energy inside the structure; $q$ is the charge; $\alpha_0$ is the pitch angle outside the structure.

If we let $\alpha_0 = 90°$, the modified critical trapping angle for an electron (i.e., $q = -e$) would be:

$$\alpha_l = \arcsin\left(\sqrt{\frac{B}{B_t} \cdot \frac{E_k - e\Phi}{E_k}}\right) \tag{7}$$

which means that it is possible for an electron to be trapped in a strong magnetic field region (i.e., $B > B_t$) if $\Phi > 0$.

### Sounding technique
The particle sounding technique can be used to determine the boundary orientation and distance to the spacecraft based on observed non-gyrotropic distributions of energetic particles which are caused by a sharp boundary (called finite Larmor radius effect) close to the spacecraft (within twice of the gyro-radius)[37]. The sounding technique has been used to determine the scale of electron structures, such as magnetic holes, and successfully reveal their geometry. Please refer to the ref. [37] for more technical details.

## Data availability
The MMS datasets during the current study are publicly available from the MMS Science Data Center (https://lasp.colorado.edu/mms/sdc/public/).

## Code availability
The MMS data are processed and analyzed using the IRFU-MATLAB package available at https://github.com/irfu/irfu-matlab.

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

## Acknowledgements

We are very grateful to the entire MMS team for providing high-resolution fields and plasma data. We thank the FGM team and the leader C.T. Russell for the DC magnetic field data, the FPI team and the leader C.J. Pollock for the electron data, and the EDP team and the leader

R.E. Ergun, P.-A. Lindqvist for the data of electric field. This work was supported by the China Space Agency projects of D020301 (Q.G.Z.) and D020303 (Q.G.Z.), the National Natural Science Foundation of China 42011530080 (Q.G.Z.) and 41974191 (Q.G.Z.), and the Science and Technology Development Fund, Macau SAR (File no. SKL-LPS(MUST)-2024-2026) (Q.G.Z.).

## Author contributions

Z.K.X. performed the data analysis, contributed to data interpretation, and prepared the paper. Q.G.Z. provided the physical idea for the overall paper. C.Y., X.Z.Z., J.S.H., Z.Y.L., and Y.X.H. contributed to data interpretation and the paper writing. H.Z., C.S.N., and S.T.Y. contributed to the revision of the paper. C.P., G.L., R.E.E., and P.A.L. contributed to the assurance of MMS data quality.

## Competing interests

The authors declare no competing interests.
