## [Peer Review File · Nature Communications]

Electron Scale Coherent Structure as Micro Accelerator in the Earth's MagnetosheathREVIEWER COMMENTS

Reviewer #1 (Remarks to the Author):

Key results

This manuscript considers plasma dynamics in an hourglass-like magnetic field configuration under the influence of a field-aligned electric field. It is shown that electron trapping and acceleration in such a configuration can explain the pitch angle distribution and bidirectional electron jets observed in the dayside magnetosheath.

Significance

Turbulent energy dissipation and plasma acceleration in electron-scale coherent structures are important topics in space plasma science. The novelty of the proposed model is the effect of the field-aligned electric field in the magnetic mirror configuration. This model helps to explain electron-scale plasma dynamics in the magnetosheath and potentially provides new types of electron accelerators and jet drivers in space plasma environments.

Data and methodology

The satellite data analysis is outside the scope of my expertise.

Analytical approach

The analytical approach is based on the standard plasma confinement model in the mirror configuration. The authors add a field-aligned electric field based on the observation results. I think this approach is valid for the present case.

Suggested improvements

The manuscript could be suitable for publication after the authors address the following concerns to clarify the significance and the explanation.

1. The present configuration is similar to the mirror mode structure, which has been widely investigated in relation to plasma turbulence, electron acceleration, and non-Maxwellian distribution in space plasmas (for example, see the references in the Introduction section by S. Yao et al., 2018). The authors should clarify the differences from these previous studies, such as the magnetic field configuration, the formation mechanism, and how the presence of the electric field modifies the previous results.

2. Could you provide the possible origin of the field-aligned electric field to ensure the generality and consistency of the present model to apply other plasma environments?
3. Line 140: More detailed explanation is needed here because the perpendicular velocity increases at the end of the magnetic mirror, where the magnetic field strength is maximum, even without the electric field.
4. The authors should address the choice of the position for zero electric potential and plasma condition there in the present model and the observation result because this could affect Eq.5 and related discussions.
5. Figure 2(d): According to Fig. 1(g), the electric potential should be non-zero for line-4.
6. Line 202: Why the electric potential can be neglected to demonstrate the electron jet structure in Fig.2 (d) at line-4?
7. Lines 263 and 284: Reference number 48 is needed for Boldyrev et al. (2020).
8. Equation 1: Is the approximation $v \sim v_{\parallel}$ applicable to the present case?
9. It seems that the potential generated by the gradient force introduced in the Discussion and Outlook section does not appear in the MMS observation in Fig. 1(g). How does this affect the interpretation of the observation?

Reviewer #2 (Remarks to the Author):

The authors of the manuscript "Electron Scale Coherent Structure as Micro Accelerator in the Earth's Magnetosheath" present an observational study of a coherent magnetic-field structure in the magnetosheath. They underpin their observational work with a model for the trapping of electrons in the magnetic and electric fields associated with this structure. This work is very interesting as it potentially points at a new type of coherent structure in space plasmas. This work is relevant to a

broader readership base and thus suitable for publication in Nature Communications. There are some major shortcomings in the presentation which mean that I cannot recommend acceptance of the manuscript without a major revision.

Major remarks:

1) The authors use $\mathbf{j} \cdot \mathbf{E}$ as a measure for dissipation. Some more care is required with this interpretation. In lines 64 and 65, the authors introduce $\mathbf{j} \cdot \mathbf{E}$ and the pressure-strain interaction as conversion processes that transfer energy from other forms into thermal energy. However, the thermal-energy equation does not include $\mathbf{j} \cdot \mathbf{E}$. In fact, $\mathbf{j} \cdot \mathbf{E}$ only contributes to the bulk kinetic energy balance. In that sense, $\mathbf{j} \cdot \mathbf{E}$ is not a direct dissipation marker (unless collisions or other actual dissipation processes are present). This issue also applies to line 135, where $\mathbf{j} \cdot \mathbf{E}$ is linked to heating.

A second comment regarding $\mathbf{j} \cdot \mathbf{E}$: the manuscript does not explain how the authors calculate $\mathbf{j} \cdot \mathbf{E}$. A sufficient level of detail is required in order to guarantee reproducibility of the results.

2) The underlying model for electron trapping in combined magnetic and electric fields is not derived with sufficient detail. In particular, it is not clear where Eq. (5) comes from. What are the key assumptions regarding adiabatic invariants? What assumptions are made on the direction of the electric field along the particle trajectory? Without a more detailed derivation, this model cannot be used as a reliable interpretation for the observations.

Minor remarks:

1) Line 52: The distinction between co-convecting and propagating coherent structures is not relevant for this manuscript. I recommend removing this sentence.

2) Line 113: The authors define the parallel field direction as the mean of field of the four MMS spacecraft. How much do the individual field measurements vary? This variability can be seen as a measure for the error of the field direction, which may impact the definitions of key quantities such as the parallel electric field.

3) Line 151: The manuscript does not provide sufficient detail about how the potential has been estimated. What defines the maximum potential difference (e.g., 50 V) in the model? How does the potential relate to measurements of E_{par} and E_{perp} ?

4) Section starting line 177: I encourage the authors to explain that the resulting distribution of trapped particles is a torus distribution as shown in Figure 2 (b).

- 5) Line 208: The reference to non-gyrotropy in Figure S1 is not clear. What do the authors refer to? Is it the gyro-phase bunching that can be seen in some panels between pitch-angles of 60 and 90 degrees? This should be explained more clearly.
- 6) Line 210: The manuscript does not provide sufficient information to understand the sounding method. This should be clearly and reproducibly described in the Methods section.
- 7) Line 241: This sentence is unclear. If the field changes the sign, it is locally equal to zero, in which case the acceleration also drops to zero.
- 8) Line 243: The last sentence of the paragraph is highly speculative and not supported by the results of this work. It is unclear what "dynamic balance" the authors refer to. I recommend removing this sentence.
- 9) Line 249: It is not clear why the authors think that the hourglass shape is the most reasonable (or even the only?) possible structure consistent with the observations. If it is only one possibility amongst many, that should be stated more clearly.
- 10) Line 260: How is the potential due to the gradient force of the magnetic field calculated? The detailed method and the relevant equations should be given in the Methods section.
- 11) Equations (1) and (2): The approximation (last part of the equation) shown in Eq. (1) is confusing. It is actually not used in the manuscript anywhere. Instead, Eq. (2) requires the full inequality without the approximation.
- 12) Paragraph starting in line 284: The authors provide an idea as to where the jet electrons come from. This may be the case for those with a pitch-angle directed away from the coherent structure. Where does the oppositely directed electron beam come from? These are electrons entering the coherent structure from the outside, so they would need to be reflected back outside the structure (if they are indeed related to the structure).
- 13) Caption of Figure S1: Define the abbreviation "DBCS".

14) Figure S5 is not referenced in the main manuscript. It also includes some quantities that have not been introduced or described in the manuscript. Should this figure simply be removed?

Reviewer #3 (Remarks to the Author):

Review Report on Ms. "Electron scale coherent structure as micro accelerator in the Earth's magnetosheath" by Xie et al.,

This work deals with the identification of a specific structure at the electron scale in turbulent space plasma capable of accelerating electrons so as to transform their distribution into a stream (Stahl) distribution.

This is evidence of a type of coherent structure in plasma turbulence at electron microscales.

I found the manuscript well conceived and quite interesting as an example of the existence of microscope electron structures. However, there are some issues that, in my opinion, do not justify the publication in Nature Communications.

First of all, although the analysis of the observed structure is well conceived and done, the authors do not explain the relevance of this structure in relation to the occurrence of turbulence and dissipation at electron scales. In their discussion, there is not a clear connection in connection with dissipation mechanisms and why these structures should be relevant for them. Indeed, the authors write: "At the end of the structure, a bidirectional electron jet is formed due to an outward parallel electric force together with the outward mirror force, which further accelerates electrons and impacts on electron dynamics in the ambient plasma." and so what?

Second, it is not clear how much statistical relevance this kind of structure has. Is this just one of the myriad of possible electron-scale structures? Or not?

Third: Although the authors state that "Intermittent coherent structures with stronger current density, especially for the first class, are usually associated with enhancements in temperature, indicating plasma heating due to dissipation of coherent structures", there is not clear evidence of intermittency at electron scales to justify that the observed structure is related to heating. The works by Osman et al. (Ref. 28) and Servidio et al. (Ref. 29) refer to scales near the ion inertial length and to structures larger than the ion-inertial length (currents are typically some ion-inertial length in thickness).

On the basis of the above comments, I believe that this manuscript is not suitable for publication in Nature Communications since I do not see any relevant advancement in understanding the physical mechanisms of dissipation in noncollisional space plasmas. Thus, I do not recommend it for publication.

Response to Reviewers

We sincerely appreciate the reviewers for their very constructive comments, which certainly provide us the opportunity to improve our manuscript. We have carefully considered all the comments and revised the manuscript accordingly. Please find below a point-to-point response to these comments, in which the reviewers' comments are shown in blue and our replies are in black.

To Reviewer #1

Key results

This manuscript considers plasma dynamics in an hourglass-like magnetic field configuration under the influence of a field-aligned electric field. It is shown that electron trapping and acceleration in such a configuration can explain the pitch angle distribution and bidirectional electron jets observed in the dayside magnetosheath.

Significance

Turbulent energy dissipation and plasma acceleration in electron-scale coherent structures are important topics in space plasma science. The novelty of the proposed model is the effect of the field-aligned electric field in the magnetic mirror configuration. This model helps to explain electron-scale plasma dynamics in the magnetosheath and potentially provides new types of electron accelerators and jet drivers in space plasma environments.

Data and methodology

The satellite data analysis is outside the scope of my expertise.

Analytical approach

The analytical approach is based on the standard plasma confinement model in the mirror configuration. The authors add a field-aligned electric field based on the observation results. I think this approach is valid for the present case.

Suggested improvements

The manuscript could be suitable for publication after the authors address the following concerns to clarify the significance and the explanation.

We are very grateful to the reviewer for the constructive comments. We have given full consideration

to the comments and revised manuscript thoroughly. Please find our detailed responses to the comments in the following letter.

1. The present configuration is similar to the mirror mode structure, which has been widely investigated in relation to plasma turbulence, electron acceleration, and non-Maxwellian distribution in space plasmas (for example, see the references in the Introduction section by S. Yao et al., 2018). The authors should clarify the differences from these previous studies, such as the magnetic field configuration, the formation mechanism, and how the presence of the electric field modifies the previous results.

Thank you for reminding us of the S. Yao et al. (2018) paper, a great supplement to our references. Mirror modes are non-propagating compressional structures frequently observed in the magnetosheath. They manifest as significant variations in the magnetic field, distinct magnetic weakenings (troughs), or magnetic enhancements (peaks), accompanied by corresponding anticorrelated features in the plasma density.

Regarding the formation mechanism, there is a strong parallel electric field in the electron scale structure reported in our manuscript. The formation mechanism of this structure and the parallel electric field may be related to the nonlinear evolution of electron holes, solitary waves, and other structures, which satisfy the self-consistent nonlinear Vlasov-Maxwell equations, in which the existence of a parallel electric field is allowed.

Unlike the mirror mode, the parallel electric field in the structure is enough to change the dynamic characteristics of electrons. We introduce the energy term caused by the electric field into the capture line of the pitch angle spectrum, and the results agree with the observations. Most importantly, we found that the parallel electric field of this structure can accelerate electrons and produce a jet, which is different from the previously observed electron-scale mirror mode, such as S. Yao et al. (2018), there is no apparent electron jet in the pitch angle distribution of mirror mode structure.

Reference:

Yao, S. T. *et al.* Electron dynamics in magnetosheath mirror-mode structures. *Journal of Geophysical Research: Space Physics* **123**, 5561-5570 (2018).

2. Could you provide the possible origin of the field-aligned electric field to ensure the generality and consistency of the present model to apply other plasma environments?

Thank you for your suggestion. We propose that the field-aligned electric field in this structure is related to the nonlinear evolution of electron holes, solitary waves and other structures satisfying the self-consistent nonlinear Vlasov equations, in which the existence of parallel electric field is allowed. In fact, we give the local electric potential linearly, and then use the electric potential and the observed magnetic field to get the local loss cone. We found that in the velocity space, the local loss cone is in good agreement with the observation results, which reflects our model correctness.

3. Line 140: More detailed explanation is needed here because the perpendicular velocity increases at the end of the magnetic mirror, where the magnetic field strength is maximum, even without the electric field.

Figure R1 | Panels (i) - (l) show pitch angle distributions of electron phase space density of energies from $\sim 90\text{eV}$ to $\sim 200\text{eV}$. The details shown in other panel are consistent with those in Figure 1 of the main text.

We fully agree with the reviewer. Our original thoughts are as follows. Provided no electric field exists here, according to Liouville's theorem, the phase space density (PSD) of 90° pitch angle electrons at the magnetic field maximum region should be equal to the PSD of $< 90^\circ$ pitch angle electrons at the lower magnetic field region. However, in the lower magnetic field region (e.g., line-

2), the PSD of $< 90^\circ$ pitch angle electrons are smaller than the 90° pitch angle PSD at the magnetic field maximum region as shown in Figure R1i-11. Therefore, the changes in PSD imply the existence of an acceleration process in this structure.

Although the parallel electric field at line-1 and line-2 is 0, the parallel electric field between line-1 and line-2 is not 0. This electric field can trap particles together with the magnetic field (from line-2 to line-3, inside the magenta line). As for the part from where the magenta line ends to line 3, it is a mixture of background and jet particles, which are undergoing the acceleration process of the electric field.

However, in this round of review, we decided to remove this sentence to any potential uncertainty and ambiguity.

4. The authors should address the choice of the position for zero electric potential and plasma condition there in the present model and the observation result because this could affect Eq.5 and related discussions.

The potential is defined to be zero at line-1 in Figure 1, following which the parallel electric field begins to deviate significantly from zero.

5. Figure 2(d): According to Fig. 1(g), the electric potential should be non-zero for line-4.

Sorry for the confusion. In original Figure 2d, the black dotted lines are derived solely from the magnetic fields. Now, as a comparison, we have added a blue trapping line that takes both the electric and magnetic fields into account. Please find them in the Figure captions and the revised Fig. 2 (d).

6. Line 202: Why the electric potential can be neglected to demonstrate the electron jet structure in Fig.2 (d) at line-4?

Sorry for the confusion. As described in our response to comment 5, the black dashed line in Figure 2 (d) shows the local loss cone without electric fields.

7. Lines 263 and 284: Reference number 48 is needed for Boldyrev et al. (2020).

We have revised the text accordingly.

8. Equation 1: Is the approximation $v \sim v_{\parallel}$ applicable to the present case?

By showing equations 1-3, which are taken from Boldyrev et al. (2020), we want to show that the hyperbolic form of the critical trapping pitch angle is theoretically expected. However, after carefully reconsideration, we now find they are not very relevant to our discussion. They are just a more sophisticated version of equations 4-5, and are not referred to in the rest of our manuscript. Therefore, we decided to remove equations 1-3 and only keep the reference Boldyrev et al. (2020).

9. It seems that the potential generated by the gradient force introduced in the Discussion and Outlook section does not appear in the MMS observation in Fig. 1(g). How does this affect the interpretation of the observation?

We sincerely appreciate the careful review. The magnetic gradient force given in the discussion part is introduced to explain the structure qualitatively. The effect of magnetic mirror force on particles

is described by the local loss cone in the pitch angle spectrum in Fig. 1(g). If we want to quantitatively calculate the magnetic field gradient potential from the observation, we need to use the multi-satellite gradient algorithm. However, this algorithm is not applicable here, since the separation among the four MMS satellites are much larger than the scale of the structure considered; this unfortunate fact would introduce significant errors in any quantitative calculation of the gradient. We emphasize in line 268 that this is a qualitative diagram.

To Reviewer #2

The authors of the manuscript "Electron Scale Coherent Structure as Micro Accelerator in the Earth's Magnetosheath" present an observational study of a coherent magnetic-field structure in the magnetosheath. They underpin their observational work with a model for the trapping of electrons in the magnetic and electric fields associated with this structure. This work is very interesting as it potentially points at a new type of coherent structure in space plasmas. This work is relevant to a broader readership base and thus suitable for publication in Nature Communications. There are some major shortcomings in the presentation which mean that I cannot recommend acceptance of the manuscript without a major revision.

We are very grateful to the reviewer for the constructive comments. We have given full consideration to the comments and revised manuscript thoroughly. Please find our detailed responses to the comments in the following letter.

Major remarks:

1) The authors use $\mathbf{j} \cdot \mathbf{E}$ as a measure for dissipation. Some more care is required with this interpretation. In lines 64 and 65, the authors introduce $\mathbf{j} \cdot \mathbf{E}$ and the pressure-strain interaction as conversion processes that transfer energy from other forms into thermal energy. However, the thermal-energy equation does not include $\mathbf{j} \cdot \mathbf{E}$. In fact, $\mathbf{j} \cdot \mathbf{E}$ only contributes to the bulk kinetic energy balance. In that sense, $\mathbf{j} \cdot \mathbf{E}$ is not a direct dissipation marker (unless collisions or other actual dissipation processes are present). This issue also applies to line 135, where $\mathbf{j} \cdot \mathbf{E}$ is linked to heating.

We sincerely appreciate this comment. As a microscopic variable, $\mathbf{J} \cdot \mathbf{E}$ is a measure of the conversion between electromagnetic field energy and particle kinetic energy, which is widely used in the study of reconnection, coherent structures and wave-particle interactions. This variable can quantitatively show whether the energy transfer is directed from fields to particles, or from particles to fields. In the former case, this variable serves as a proxy of the dissipation of field energy. Of course, we agree with the reviewer that one cannot directly figure out energy dissipation mechanism from $\mathbf{J} \cdot \mathbf{E}$; other investigation on micro-physics is required. In our manuscript, we proposed such a microscopic mechanism by examining simultaneously the field and particle observations.

A second comment regarding $\mathbf{j} \cdot \mathbf{E}$: the manuscript does not explain how the authors calculate $\mathbf{j} \cdot \mathbf{E}$. A sufficient level of detail is required in order to guarantee reproducibility of the results.

Thanks very much for identifying this issue. We are sorry for the lack of detailed information of calculating $\mathbf{j} \cdot \mathbf{E}$. Please find these revisions in lines 135-136 and below:

1. We only used data from MMS1 to calculate $\mathbf{J} \cdot \mathbf{E}$.
2. For the magnetic field \mathbf{B}_{gse} in the GSE coordinate system, we interpolate the \mathbf{B}_{gse} to the electron sampling time of FPI-DES.
3. The ion velocity of FPI-DIS \mathbf{V}_i is interpolated to the electron sampling time of FPI-DES.
4. Assuming $N_i=N_e$, we use N_e to calculate the current in the GSE coordinates, that is, $\mathbf{J}_{gse} = e \cdot$

$\mathbf{Ne} \cdot (\mathbf{Vi} - \mathbf{Ve})$.

5. For the electric field in the GSE coordinate system, we first applied a running average on \mathbf{Egse} with the window set as the FPI electron sampling interval, and then interpolated the \mathbf{Egse} to the FPI electron measurement time.

6. Then we calculate the electric field in the electron frame, $\mathbf{E}' = \mathbf{Egse} + \mathbf{Ve} \times \mathbf{Bgse}$.

2) The underlying model for electron trapping in combined magnetic and electric fields is not derived with sufficient detail. In particular, it is not clear where Eq. (5) comes from. What are the key assumptions regarding adiabatic invariants? What assumptions are made on the direction of the electric field along the particle trajectory? Without a more detailed derivation, this model cannot be used as a reliable interpretation for the observations.

Thanks very much for the comments, and sorry that we missed detailed derivation. Please find these revisions in lines 320-333 and below:

In the potential field induced by the parallel electric force, the particle's total energy W remains constant. Recalling $\Phi(s)$ denoting the potential, we have:

$$W = E_k(s) + q\Phi(s) = E_{k0} + q\Phi_0 = \text{const} \quad (1)$$

Assume that the initial potential is 0 (The 0 subscript represents the area with no electric field), $\Phi_0 = 0$, we can get:

$$E_{k0} = E_k(s) + q\Phi(s) \quad (2)$$

We next assume that the first adiabatic invariant M is conserved:

$$M = \frac{E_{k\perp}}{B} = \frac{E_k \sin^2(\alpha)}{B} = \text{const} \quad (3)$$

Assuming that the background magnetic field strength outside the structure is B_t and substituting Eq. (2) into (3), we have,

$$M = \frac{E_k \sin^2(\alpha)}{B} = \frac{E_{k0} \sin^2(\alpha_0)}{B_t} = \frac{(E_k + q\Phi) \sin^2(\alpha_0)}{B_t} \quad (4)$$

Accordingly, the pitch angle α of a particle entered into the structure should be:

$$\sin(\alpha) = \sqrt{\frac{B}{B_t} \cdot \frac{E_k + q\Phi}{E_k}} \sin(\alpha_0) \quad (6)$$

Minor remarks:

1) Line 52: The distinction between co-convecting and propagating coherent structures is not relevant for this manuscript. I recommend removing this sentence.

We agree with the reviewer. We have removed this sentence.

2) Line 113: The authors define the parallel field direction as the mean of field of the four MMS

spacecraft. How much do the individual field measurements vary? This variability can be seen as a measure for the error of the field direction, which may impact the definitions of key quantities such as the parallel electric field.

Sorry for the confusion. We did not use measurements from four satellites to calculate the parallel electric fields. For calculating the parallel electric field, we only used the MMS1 data. The following is how we calculate the parallel electric field:

1. For the magnetic field \mathbf{Bgse} in the GSE coordinate system, we interpolate the \mathbf{Bgse} to the electron sampling time of FPI.
2. For the electric field \mathbf{Egse} in the GSE coordinate system, we first apply a running average on \mathbf{Egse} using the FPI electron sampling interval as the window and then interpolate the \mathbf{Egse} to the FPI electron measurement time.
3. Calculate the parallel electric field, $E_{para} = \mathbf{Egse} \cdot \mathbf{Bgse} / |\mathbf{Bgse}|$, the time resolution of the parallel electric field E_{para} is the same as that of the FPI's electron measurements.

We added a sentence “(Only MMS1 observations are used)” to avoid confusion. Please find them in lines 128-129.

3) Line 151: The manuscript does not provide sufficient detail about how the potential has been estimated. What defines the maximum potential difference (e.g., 50 V) in the model? How does the potential relate to measurements of E_{para} and E_{perp} ?

We sincerely appreciate the careful review. The coincidence between the trapping line after considering the electric potential and the actual electron PAD observation is used to define the electric potential indirectly. Considering that the electric potential must be continuous in space and the sign reversal of the parallel electric field occurs at line-2, the electric potential increases linearly between line-1 and line-2, then decreases linearly between line-2 and line-3, and then increases linearly between line-3 and line-4. The critical trapping line in phase space drawn by the maximum potential of 50V is in good agreement with the FPI observations, so we think this is a reasonable potential value. The electric potential of 50V mainly comes from the contribution of the parallel electric field. On the other hand, the potential changes induced by the perpendicular electric field cannot be neglected for the following reasons.

As shown in Figure R1h, the maximum amplitude of the perpendicular electric field E'_x and E'_y in the electron coordinate is about 2mV/m. Electrons considered here generally have an energy of ~100 eV, which corresponds to a cyclotron radius of about 960m. Thus, the changes in the perpendicular potential experienced by the electrons during their gyration motion is only $960\text{m} \times 2\text{mV/m} = 1.92\text{V}$, much smaller than our estimated parallel potential variations (50V). Hence, we have not considered the perpendicular electric field here.

Figure R1 (f) The electric field (E_{para}' in red, E_{perp}' in blue) in the electron coordinate. (g) The parallel electric field. (h) The perpendicular electric field in the electron coordinate. The details shown in other panel are consistent with those in Figure 2 of the main text.

4) Section starting line 177: I encourage the authors to explain that the resulting distribution of trapped particles is a torus distribution as shown in Figure 2 (b).

We are grateful for the suggestions. We further explain the distribution of trapped particles, please find them in lines 194-197 and below:

In Figure 2b, the closed region surrounded by the magenta curve represents the particles trapped by the magnetic and parallel electric fields. In contrast, outside the closed region, it means the

composition of the particles passing through freely.

5) Line 208: The reference to non-gyrotopropy in Figure S1 is not clear. What do the authors refer to? Is it the gyro-phase bunching that can be seen in some panels between pitch-angles of 60 and 90 degrees? This should be explained more clearly.

Sorry for the confusion. On each panel of Figure S1, the dashed line represents 90 pitch angles in the sky-map, while the solid line is for 60 pitch angles. Each point on these lines has a gyration phase. That is to say, at a given pitch angle, when the gyration phase changes from 0 to 360 degrees, such a trajectory will be swept out on the sky-map. We call “non-gyrotopropy” the obvious non-uniform distribution of PSD along this trajectory.

As mentioned in H. Liu (2019), for some look direction in the magnetic perpendicular plane, if the corresponding particle gyro-orbit intersects the boundary, the particle flux received by the detector in this direction will be significantly reduced due to the scattering at the boundary. Thus, the measured perpendicular PSD (PSD_{\perp}) will be non-gyrotopropic, and two critical look directions can be recognized by sharp reductions, which correspond to two special gyro-orbits that are just tangential to the boundary.

Reference:

Liu, H., Zong, QG., Zhang, H. et al. MMS observations of electron scale magnetic cavity embedded in proton scale magnetic cavity. Nat Commun 10, 1040 (2019).

6) Line 210: The manuscript does not provide sufficient information to understand the sounding method. This should be clearly and reproducibly described in the Methods section.

Thanks very much for the suggestions. We have added a new subsection titled “Sounding technique” in the Methods. Please find them in lines 344-350.

7) Line 241: This sentence is unclear. If the field changes the sign, it is locally equal to zero, in which case the acceleration also drops to zero.

Thanks for identifying this issue. We have revised this sentence to “The electrons might be accelerated in the middle of the structure by the E_{\parallel} ”.

8) Line 243: The last sentence of the paragraph is highly speculative and not supported by the results of this work. It is unclear what "dynamic balance" the authors refer to. I recommend removing this sentence.

We agree with the reviewer. Please find lines 249-250 in the revised manuscript.

9) Line 249: It is not clear why the authors think that the hourglass shape is the most reasonable (or even the only?) possible structure consistent with the observations. If it is only one possibility amongst many, that should be stated more clearly.

Thanks very much for the suggestion. We suggest an hourglass shape according to the results from the sounding technique. As shown in Figure 3c, the structure's scale is large at the ends and small in the middle, which is very similar to an hourglass.

We have added a sentence, “As shown in Figure 3c, the structure scale is large at both ends and

small in the middle, which may mean the structure is most likely to resemble an hourglass shape” to clarify our statement. Please find them in lines 254-256 in the discussion.

10) Line 260: How is the potential due to the gradient force of the magnetic field calculated? The detailed method and the relevant equations should be given in the Methods section.

Sorry for the confusion. The magnetic gradient force given in the discussion part is introduced to explain the structure qualitatively. We emphasize in line 268 that this is a qualitative diagram.

11) Equations (1) and (2): The approximation (last part of the equation) shown in Eq. (1) is confusing. It is actually not used in the manuscript anywhere. Instead, Eq. (2) requires the full inequality without the approximation.

We agree with the reviewer. We have carefully checked Equations 1 to 3. These proven equations come from Boldyrev et al. (2020), and they are not very relevant to the discussion and actually are not quoted in the rest of the text. What Equations 1 to 3 want to explain coincides with Equations 4 to 5. We want to show that the hyperbolic form of the critical trapping pitch angle is theoretically supported. Therefore, in accordance with the above argument, we decided to delete Equations 1 to 3 and keep only references.

12) Paragraph starting in line 284: The authors provide an idea as to where the jet electrons come from. This may be the case for those with a pitch-angle directed away from the coherent structure. Where does the oppositely directed electron beam come from? These are electrons entering the coherent structure from the outside, so they would need to be reflected back outside the structure (if they are indeed related to the structure).

We sincerely appreciate the careful review. Currently, we have no idea for the origination of the oppositely directed electron beam. We propose the following two hypotheses:

1. A magnetic field peak outside the structure would reflect the jet electrons coming from the structure, resulting in oppositely directed electron beam. As a piece of preliminary evidence, we show an extended time interval in Figure R2. One can see the presence of such magnetic field peaks behind the structure considered.
2. The fact that this oppositely directed electron beam mainly appear in the low energy range (<100 eV) indicates that they may just a background population not associated with the structure.

Figure R3 Panels (a) and (b) show magnetic field components (B_x in red, B_y in green, B_z in blue, and the field strength B_t in black) in the newly defined local field-aligned coordinates (detailed in the main text). Panels (i) - (l) show pitch angle distributions of electron energy flux of energies from ~ 90 eV to ~ 200 eV.

13) Caption of Figure S1: Define the abbreviation "DBCS".

The description has been added. Please find in line 7 in the Supplementary Materials.

14) Figure S5 is not referenced in the main manuscript. It also includes some quantities that have not been introduced or described in the manuscript. Should this figure simply be removed?

Figure 5 depicts the PVI results of the structure and the corresponding turbulence spectrum. We agree with the reviewer and have removed Figure S5.

To Reviewer #3

Review Report on Ms. "Electron scale coherent structure as micro accelerator in the Earth's magnetosheath" by Xie et al.,

This work deals with the identification of a specific structure at the electron scale in turbulent space plasma capable of accelerating electrons so as to transform their distribution into a stream (Stahl) distribution.

This is evidence of a type of coherent structure in plasma turbulence at electron microscales.

I found the manuscript well conceived and quite interesting as an example of the existence of microscope electron structures. However, there are some issues that, in my opinion, do not justify the publication in Nature Communications.

First of all, although the analysis of the observed structure is well conceived and done, the authors do not explain the relevance of this structure in relation to the occurrence of turbulence and dissipation at electron scales. In their discussion, there is not a clear connection in connection with dissipation mechanisms and why these structures should be relevant for them. Indeed, the authors write: "At the end of the structure, a bidirectional electron jet is formed due to an outward parallel electric force together with the outward mirror force, which further accelerates electrons and impacts on electron dynamics in the ambient plasma." and so what?

We are very grateful to the reviewer for the constructive comments and the judgment of the importance of our manuscript's topic. As mentioned by Alexandrova et al. (2012), "it is still not clear whether we can describe turbulence in the solar wind as a mixture of linear waves (weak turbulence) which will dissipate homogeneously in space (or in the plane perpendicular to B), or if it is a strong turbulence with dissipation restricted to intermittent coherent structures. What is the topology of these structures—current sheets, shocks, or coherent vortices?" A mechanism is proposed in our manuscript. The structure we discovered can dissipate energy on the electron scale. This provides a possible specific channel for turbulent dissipation. However, it is still a problem to explain how to form this structure.

Reference:

Alexandrova, O., Lacombe, C., Mangeney, A., Grappin, R., & Maksimovic, M. (2012). Solar wind turbulent spectrum at plasma kinetic scales. *The Astrophysical Journal*, 760(2), 121.

Second, it is not clear how much statistical relevance this kind of structure has. Is this just one of the myriad of possible electron-scale structures? Or not?

Thanks very much for the comments. This structure is indeed a relatively common electron-scale structure and is (at least) one of the important channels for the transmission of turbulent energy to particles. Because the interval itself accounts for a small proportion (less than 4%, Wu et al. (2023)), but it plays an important role in energy dissipation. We find many similar coherent structures in burst mode data of MMS1, and from September 1 to 7, 54 cases have been found. The corresponding

time of the structures are recorded in Table R1. However, because this structure is complex, it is not easy to identify through the program algorithm, so we do not have extensive statistics on the structure. But what we can confirm is that this is a common electron scale structure.

Reference:

Wu, H., Huang, S., Wang, X., Yuan, Z., He, J., & Yang, L. (2023). Intermittency of Magnetic Discontinuities in the Near-Sun Solar Wind Turbulence. *The Astrophysical Journal Letters*, 947(2), L22.

Third: Although the authors state that "Intermittent coherent structures with stronger current density, especially for the first class, are usually associated with enhancements in temperature, indicating plasma heating due to dissipation of coherent structures", there is not clear evidence of intermittency at electron scales to justify that the observed structure is related to heating. The works by Osman et al. (Ref. 28) and Servidio et al. (Ref. 29) refer to scales near the ion inertial length and to structures larger than the ion-inertial length (currents are typically some ion-inertial length in thickness).

Thanks very much for the comments. Results in the solar wind (Sioulas et al. (2022)) show that intermittency is related to ion heating, and the results in the context of magnetosheath (Chasapis et al. (2015)) show that the intermittence is highly related to electron heating. Most of the turbulent energy is handed over to the ions in the dissipation process, and the residual energy comes to the electron scale, which is likely to be dissipated by this structure in our manuscript.

Reference:

Sioulas, N., Shi, C., Huang, Z., & Velli, M. (2022). Preferential heating of protons over electrons from coherent structures during the first perihelion of the Parker solar probe. *The Astrophysical Journal Letters*, 935(2), L29.

Chasapis, A., Retinò, A., Sahraoui, F., Vaivads, A., Khotyaintsev, Y. V., Sundkvist, D., ... & Canu, P. (2015). Thin current sheets and associated electron heating in turbulent space plasma. *The Astrophysical Journal Letters*, 804(1), L1.

On the basis of the above comments, I believe that this manuscript is not suitable for publication in *Nature Communications* since I do not see any relevant advancement in understanding the physical mechanisms of dissipation in noncollisional space plasmas. Thus, I do not recommend it for publication.

Again, we sincerely appreciate the referee for spending time reviewing this manuscript. Respectfully, we still suggest that the electron scale structure in our study is very important. We concentrate on the key role played by the parallel electric field in this electron-scale structure in a turbulent environment, which provides a possible energy transfer channel for turbulent dissipation. Therefore, we would like to respectfully ask if the reviewer could reconsider this judgement?

Table R1 | Start time of the similar electron scale structures, observed by MMS1.

2015-09-02T13:53:34UT
2015-09-02T15:27:24UT
2015-09-02T15:27:34UT
2015-09-02T15:27:44UT
2015-09-02T15:27:54UT
2015-09-02T15:28:44UT
2015-09-02T15:29:44UT
2015-09-02T15:29:54UT
2015-09-02T16:07:34UT
2015-09-02T16:11:04UT
2015-09-02T16:12:04UT
2015-09-02T16:13:24UT
2015-09-02T16:47:44UT
2015-09-02T16:56:14UT
2015-09-02T16:56:34UT
2015-09-02T17:00:34UT
2015-09-02T17:02:14UT
2015-09-02T17:02:24UT
2015-09-02T17:02:44UT
2015-09-02T17:05:44UT
2015-09-02T17:13:54UT

2015-09-02T17:15:14UT
2015-09-02T17:16:24UT
2015-09-02T17:23:54UT
2015-09-02T17:30:24UT
2015-09-02T17:31:44UT
2015-09-03T14:24:24UT
2015-09-03T14:24:44UT
2015-09-03T14:31:04UT
2015-09-03T15:15:04UT
2015-09-03T16:14:44UT
2015-09-03T16:50:04UT
2015-09-03T16:52:54UT
2015-09-03T16:53:14UT
2015-09-03T16:54:04UT
2015-09-03T16:54:14UT
2015-09-03T16:54:24UT
2015-09-03T16:54:34UT
2015-09-03T16:54:34UT
2015-09-03T16:55:24UT
2015-09-03T17:21:04UT
2015-09-03T17:21:54UT
2015-09-03T17:22:04UT

2015-09-03T17:22:24UT
2015-09-03T17:23:04UT
2015-09-03T17:23:14UT
2015-09-07T13:32:44UT
2015-09-07T13:33:24UT
2015-09-07T13:36:44UT
2015-09-07T13:38:24UT
2015-09-07T13:57:14UT
2015-09-07T13:58:44UT
2015-09-07T13:59:04UT
2015-09-07T13:59:54UT

REVIEWER COMMENTS

Reviewer #1 (Remarks to the Author):

The authors have adequately answered my concerns.

I believe the paper is now acceptable for publication.

Reviewer #2 (Remarks to the Author):

The authors have largely revised the manuscript according to my comments. I still disagree with the identification of $\mathbf{j} \cdot \mathbf{E}$ with "Joule heating" though. Joule heating requires collisions to transfer the differential bulk velocities between ions and electrons into thermal energy. This is not the case here. I urge the authors to replace all occurrences of the term "Joule heating" with "energy transfer between fields and particles" as phrased in the response. Also, any references to "negative heating" are nonsensical.

Once this point has been clarified, the manuscript is suitable for acceptance in Nature Communications.

Reviewer #3 (Remarks to the Author):

Review Report on Ms. "Electron scale coherent structure as micro accelerator in the Earth's magnetosheath" by Xie et al.

In this new version of the manuscript, the Authors have considered all my comments from the first iteration. The manuscript has surely been improved, and they have provided evidence for more extensive statistics of the claimed structures. What convinces me less is the association of these structures with electron heating. Surely these structures can accelerate electrons, as shown by the analyses and the theoretical study, but heating remains, to me, more obscure. However, I find the new version of this manuscript suitable for publication in Nature Communication in its present form. My only requirement is that the Authors will include the list of the event reported in R1 Table as an additional material to the manuscript, by explicitly mentioning it in the text. Thus, I support the acceptance of it for publication with this minor correction.

Response to Reviewers

We are sincerely thankful for the positive comments provided by the reviewers. We have carefully considered all the comments and revised the manuscript accordingly. Please find below a point-to-point response to these comments, in which the reviewers' comments are shown in blue, and our replies are in black.

Response to Reviewer #1:

The authors have adequately answered my concerns.

I believe the paper is now acceptable for publication.

We are very grateful to reviewer #1 for his/her continued efforts in evaluating this paper.

Response to Reviewer #2:

The authors have largely revised the manuscript according to my comments. I still disagree with the identification of $j \cdot E$ with "Joule heating" though. Joule heating requires collisions to transfer the differential bulk velocities between ions and electrons into thermal energy. This is not the case here. I urge the authors to replace all occurrences of the term "Joule heating" with "energy transfer between fields and particles" as phrased in the response. Also, any references to "negative heating" are non-sensical.

We sincerely appreciate the reviewer for identifying the issues of our statements about heating. In the updated manuscript, we replace all "Joule heating" with "energy transfer between fields and particles". Also, we removed all references to "negative heating". Please see lines 131-132, 134-136, 244-245, 485-486. Please note that the line numbers are for the text with track changes.

Once this point has been clarified, the manuscript is suitable for acceptance in Nature Communications.

Once again, we sincerely appreciate the reviewer's effort in reviewing this manuscript.

Response to Reviewer #3:

Review Report on Ms. "Electron scale coherent structure as micro accelerator in the Earth's magnetosheath" by Xie et al.

In this new version of the manuscript, the Authors have considered all my comments from the first iteration. The manuscript has surely been improved, and they have provided evidence for more extensive statistics of the claimed structures. What convinces me less is the association of these structures with electron heating. Surely these structures can accelerate electrons, as shown by the

analyses and the theoretical study, but heating remains, to me, more obscure. However, I find the new version of this manuscript suitable for publication in Nature Communication in its present form. My only requirement is that the Authors will include the list of the event reported in R1 Table as an additional material to the manuscript, by explicitly mentioning it in the text. Thus, I support the acceptance of it for publication with this minor correction.

We are very grateful to the reviewer for his/her efforts in evaluating this paper. We have carefully considered the constructive comments and revised the manuscript accordingly. We notice the "electron heating" may not be appropriate as also pointed by reviewer #2. Thus, we have replaced all the "heating" throughout the paper with "energy transfer between fields and particles" to avoid confusion. Please see lines 131-132, 134-136, 244-245, 485-486. Please note that the line numbers are for the text with track changes.

In addition, we have included the list of the events reported in R1 Table in Supplementary Table 1, and explicitly mention it in lines 288-289:

“More similar coherent structures can be found in Supplementary Table 1.”

Once again, we sincerely appreciate the reviewer's effort in reviewing this manuscript.

REVIEWERS' COMMENTS

Reviewer #3 (Remarks to the Author):

Review Report on Ms. "Electron scale coherent structure as micro accelerator in the Earth's magnetosheath" by Xie et al.

This new version of the manuscript has practically unchanged in respect to the previous one, apart from some sentences where the term "Joule heating" is modified with "energy transfer".

Thus, my position remains the previous one, i.e., the paper is interesting and the results too, so that it can be accepted for publication in Nature Communications.

There is, however, one point that it is not clear to me and that the Editor should consider before accepting the manuscript for publication.

THE LIST OF THE AUTHORS IS CHANGED. S. BOLDYREV is no longer present in this paper.

I do not know if this is possible during the reviewing procedure. Thus, I leave this point to the Editor decision.

Response to Reviewers

First of all, we are grateful to the reviewer #3 for this constructive comment, which certainly provides us the opportunity to improve our paper. We have carefully considered the comment and accordingly revised the manuscript. Please find below a point-to-point response to these comments, in which the reviewers' comments are shown in blue, and our replies are in black.

Response to Reviewer #3:

Review Report on Ms. "Electron scale coherent structure as micro accelerator in the Earth's magnetosheath" by Xie et al.

This new version of the manuscript has practically unchanged in respect to the previous one, apart from some sentences where the term "Joule heating" is modified with "energy transfer".

Thus, my position remains the previous one, i.e., the paper is interesting and the results too, so that it can be accepted for publication in Nature Communications.

There is, however, one point that it is not clear to me and that the Editor should consider before accepting the manuscript for publication.

THE LIST OF THE AUTHORS IS CHANGED. S. BOLDYREV is no longer present in this paper.

I do not know if this is possible during the reviewing procedure. Thus, I leave this point to the Editor decision.

We are very grateful to reviewer #3 for his/her continued efforts in evaluating this paper.